# Life Story Book to enhance communication in persons with dementia: A systematic review of reviews

**Ponnusamy Subramaniam[1,2], Preyaangka Thillainathan[2,3], Nur Amirah Mat Ghani[3], Shobha Sharma [ID][2,3]***

**1** Clinical Psychology & Behavioural Health Programme, Faculty of Health Sciences, Universiti Kebangsaan Malaysia, Kuala Lumpur, Malaysia, **2** Centre for Healthy Ageing and Wellness, Faculty of Health Sciences, Universiti Kebangsaan Malaysia, Kuala Lumpur, Malaysia, **3** Speech Sciences Programme, Faculty of Health Sciences, Universiti Kebangsaan Malaysia, Kuala Lumpur, Malaysia

* shobha.sharma@ukm.edu.my

**Data Availability Statement:** All relevant data are within the paper and its Supporting information files.

## Abstract

The Life Story Book has been commonly used in promoting person-centred care in older adults, especially for persons with dementia. This involves collecting the life stories and memories of the person living with dementia and compiling them into a book or folder, which is used by staff or family to assist the person recall these memories. Evidence on the use, benefits and influences of the Life Story Book in dementia care is limited. This systematic literature review aimed to collect past reviews and provide a thorough overview of the use, benefits, and impact of the Life Story Book for the person with dementia, the relatives, family, and caregivers. The electronic databases PubMed, Scopus, Science Direct and Web of Science as well as grey literature through Google Scholar were searched to select the relevant studies. Seven studies that meet the inclusion criteria were selected and data synthesised. Findings revealed that the use of the Life Story Book has no specific guidelines and has been described with numerous characteristics and varied implementation methods. The Life Story Book intervention is found to provide positive outcomes for the person with dementia and the carers involved. Six out of the seven studies reported that Life Story Book enhanced communication between persons with dementia, relatives, care staff, and residents. The review extends the current evidence on the usage of the Life Story Book in dementia care and confirms that the use of life stories leads to better care in various settings. However, more research is needed to reveal the potential of the Life Story Book in enhancing communication. Guidelines and training are also required to make the best use of the Life Story Book.

## Introduction

Dementia is an umbrella term used for a group of diseases characterised by a progressive decline in global function including cognition, social functioning as well as behavioural changes occurring in a state of clear consciousness [1, 2]. The prevalence of dementia has

**Funding:** No funding was received for this research.

**Competing interests:** The authors have declared that no competing interests exist.

increased with the fast-growing ageing population, where it is estimated to currently affect over 50 million people globally and is estimated to increase to 152 million by 2050 [3].

A person with dementia generally experiences a gradual decline of a wide range of abilities over the years and a gradual loss of independence in daily function [4]. This loss of function requires reliance on a caregiver to fulfil their needs. The increasing number of persons with dementia has raised the demand for the provision of care. Treatment of persons with dementia is through multifactorial management, which can be categorised into pharmacological and non-pharmacological treatments [5]. The current pharmacological treatments for a person with dementia are only symptomatic and are administered by medication that has been approved by the American Food and Drugs Administration (FDA) [6]. However, medications only ameliorate the conditions and frequently provide only moderate desired outcomes in terms of managing the symptoms. Non-pharmacological treatments are recommended by experts as the first line of treatment to help the person with dementia [7] to both improve the condition of dementia and the quality of life [8] and their caregivers. Non-pharmacological treatments include cognitive training and rehabilitation, music-based therapy, light therapy, psychological therapy, cognitive stimulation therapy and reminiscence therapy [9].

Reminiscence therapy is an established non-pharmacological treatment option in dementia care that involves activities recalling past events and experiences of the person with dementia, aided by familiar items, artefacts, photographs, music, and songs that help to trigger the memory [10, 11]. The usage of the Life Story Book is one of the approaches of reminiscence work. Based on McKeown et al. [12], the Life Story Book is described as an approach that involves the person with dementia and/or their family to look at their life and history, collecting and compiling the information relating to the person with dementia and then using the information in the care of the person. The creation of a Life Story Book involves discussing the individual's past and present life using related images, sentences or memorabilia that benefit the individual to recall and focus on a specific segment of life which is later verbalised in the context of guided communication [10]. The Life Story Book which is commonly used in clinical practice in many countries [13], outlines a collection of photographs and written captions regarding the past and present story of the person with dementia and is often collated in a chronological format [14, 15].

The major focus in the creation of the Life Story Book is to facilitate person-centred care by following the main principles and philosophy of its development, which is to maintain and promote personhood in dementia care [16]. In recent years, the person-centred care approach has been widely implemented by practitioners and nurses who are working with persons with dementia [10]. The key elements of the person-centred care approach include valuing the person as they are, understanding the biography of the person and developing relationships between family, relatives, and caregivers [17, 18]. There is some evidence that indicates that the experience of creating a Life Story Book is immensely beneficial in improving communication, promoting better relationships between family, friends, and caregivers, as well as more person-centred care for persons living with dementia [19].

Language and communication problems from the deterioration of cognitive functioning are commonly experienced by a significant number of persons with dementia that necessitates professional assistance [20, 21]. This impacts the person with dementia's ability to express his/her daily needs and thoughts to family, friends, and caregivers, which in turn reduces self-esteem and quality of life [22]. The role of Speech-Language Pathologists (SLPs) is crucial in managing and caring for persons with dementia. They are required to assess, diagnose, and provide the best intervention for persons with dementia that presents with language, communication, and cognitive deficits [23]. The Life Story Book is known to benefit persons with dementia by improving their communication, memory, and cognitive functions [10]. SLPs

could therefore use the Life Story Book as a memory cue and as an intervention strategy to improve communication, language, and cognitive skills among persons with dementia [24].

A few systematic reviews have been published which have discussed the usage of the Life Story Book among persons with dementia and evidenced the effectiveness of the Life Story Book in treating persons with dementia. Although there were no standard protocols reported in the systematic reviews in terms of the protocols of conducting this intervention, the general findings reported that Life Story Book effectively improves the quality of life and other outcomes for the person living with dementia. Benefits may include improved cognition and mood, communication, autobiographical memory, and social interactions and reduced depression. As such, the objective of this paper was to perform a systematic review of the usage of the Life Story Book among persons with dementia. This paper aims to present a thorough synthesis of available systematic review literature concerning the advantages of using Life Story Book for the person with dementia, the standard procedure in preparation and usage of the Life Story Book and the impact of the use of Life Story Book on families, relatives, and caregivers. It systematically reviews past reviews on how Life Story Book is used in dementia care with three guiding questions, following the PICO model framework (Population, Intervention, Comparison and Outcomes) [25] as below:

1. What are the characteristics and standard protocols in using the Life Story Book for persons with dementia?

2. What are the benefits of the Life Story Book for the person with dementia?

3. What is the impact of using the Life Story Book on the family, relatives, and caregivers of a person with dementia?

## Method

This systematic review was conducted according to the Preferred Reporting Items for Systematic Reviews and Meta-Analyses (PRISMA) [26]. A systematic review of reviews is commonly used to provide a summary of secondary literature that includes many primary studies in the area and includes interventions from various levels, age populations, and settings [27]. In this paper, separate reviews were compared, contrasted, and synthesised into a thorough overview of a large body of work. All methods were performed according to relevant guidelines.

### Inclusion and exclusion criteria

The eligibility criteria for study inclusion were developed using PICOS as a guideline (Table 1).

### Search methods for identification of reviews

A literature search was performed in four electronic databases: PubMed, Scopus, Science Direct and Web of Science on 26th August 2022. The grey literature for relevant studies was also conducted through Google Scholar. The key search terms included were 'life story book' OR 'life story' OR 'life story work' OR 'memory album' and in combination with the Boolean operator 'AND' along with the search terms 'dementia' AND 'Alzheimers*' AND 'review'. The reference list of selected articles was also searched. The searches were limited to the English language only. The results of the searches were extracted into Microsoft Excel Worksheet and de-duplicated manually by the researcher. Eligibility assessment was performed by PT and PS independently and any disagreement between reviewers was resolved by SS's consensus.

Table 1. Eligibility criteria for study inclusion.

|  | Criteria |
|---|---|
| Population | Reviews of studies were limited to adult persons with dementia, across all stages of dementia (e.g. mild—severe), and in any setting. |
| Intervention | The review of studies of the Life Story Book: All forms of the Life Story Book (e.g., tangible book, multimedia, memory album) were included. For this study, the Life Story Book was defined as any process that involves working with a person and/or their family to collect and record the information and/or stories about the person's past and present life to be used to benefit in their care [28], that is often supported with photographs, text, or memorabilia, relevant to the person's life [13]. |
| Comparator | This review did not intend to compare the Life Story Book to/with any other forms or types of intervention. |
| Outcome | To include the study in this review, the study outcomes should have reported the advantage of using the Life Story Book among persons with dementia, and/or present the guidelines for using the Life Story Book, and/or discuss the impact of using the Life Story Book for the caregiver/relatives. |
| Study Type | Any quantitative and qualitative review paper was included (systematic review, scoping review, or narrative review). |

## Study selection

The title and abstracts of all articles obtained from the electronic search were screened by two researchers independently. Any article that was potentially eligible by either researcher was accepted for the next round of evaluation. Acquisition of full texts of articles with inadequate information in the title and abstracts allowed for precise screening. Full texts of all articles selected following the screening rounds were then obtained. These full-text articles underwent an assessment process using the inclusion criteria (Table 1). Articles that met the inclusion criteria were then evaluated for methodological quality assessment. A third researcher reviewed disagreements (SS) where a consensus could not be reached between the researchers.

## Quality assessment

Studies included were evaluated using the Measurement Tool to Assess Systematic Review (AMSTAR) checklist for assessing methodological quality [29]. The AMSTAR is a validated tool to assess the methodological quality of systematic reviews, which contains 11 items. The final grading of each systematic review was based on the overall score (total score: 11) and was reported as 'high' (score$\geq$8), 'medium' (score 4–7) or 'low' (score$\leq$3). For narrative review articles included, the methodological quality was assessed based on Enhancing Transparency in Reporting the Synthesis of Qualitative Research (ENTREQ) statement guidelines. ENTREQ summarised statement consists of 11 items grouped into five main domains: introduction, methods and methodology, literature search and selection, appraisal, and synthesis of findings. The quality of assessment of each paper was completed by NAMG and PT independently, and crossed-checked by PS.

## Data extraction

From each of the included studies, the following data were collected: the study characteristics, the publication year, the years and number of the primary studies included, the databases searched, the study population and setting, the outcome measures and the AMSTAR score for quantitative and ENTREQ evaluation for qualitative review articles. An Excel spreadsheet was created for data extraction purposes. NAMG and PT conducted data extraction independently and the data extraction was verified by PS.

## Results

### Study identification and selection

The PRISMA flow diagram in Fig 1, provides a concise overview of the exclusion and inclusion of database results leading to the final selection. A total of 119 studies were found: PubMed (7), Scopus (6), Science Direct (12), Web of Science (16), and Google Scholar (78). After removing the duplicates, 96 papers remained and underwent a second round of screening for title and abstract. Based on the title and abstract, studies were removed as the studies did not relate to Life Story Book in dementia care. The full texts of the remaining 17 studies were assessed for the inclusion criteria and 10 studies were discarded as they did not meet the inclusion criteria. Finally, a total of seven studies met the inclusion criteria and were included in this systematic review.

### Study characteristics

The seven review studies included were published between 2006 to 2020. All papers were written in English. Table 2 depicts the basic characteristics of the included reviews. Out of these seven studies, five studies were systematic reviews [10, 13, 30–32] and two studies were qualitative review studies [14, 33]. The population included in the primary studies were persons with

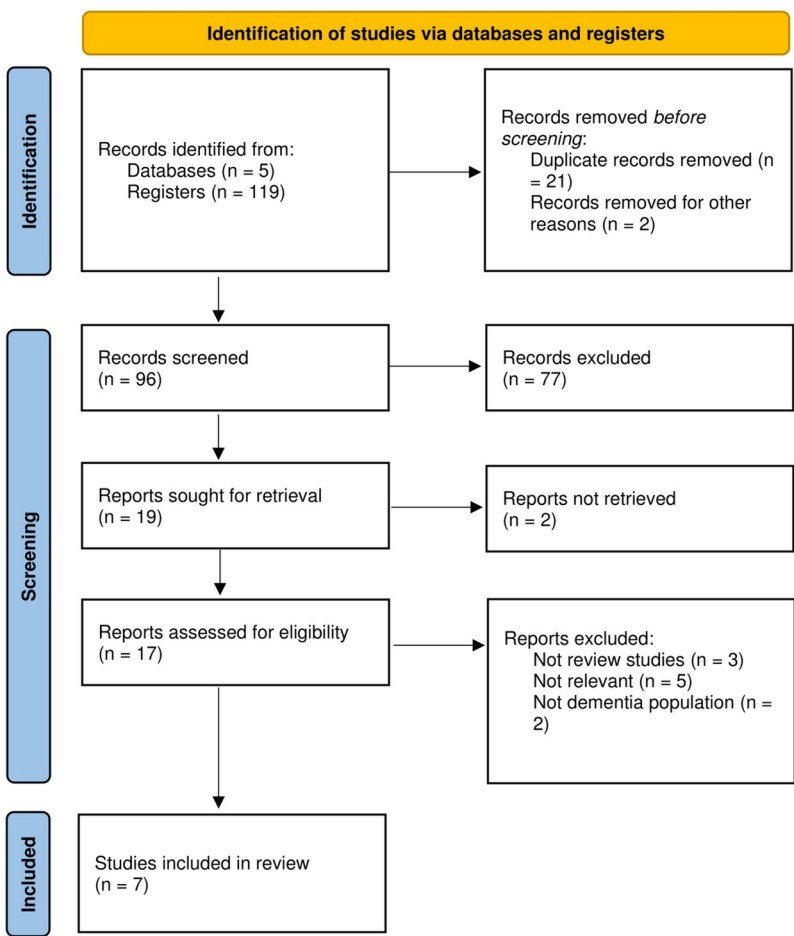

**Fig 1. Flow diagram for Life Story Book review articles for persons with dementia.**

**Table 2. Overview and characteristics of the included studies.**

| No | Authors / Year | Aim | Setting | Year of primary studies included | Study design of included studies | Population (person with dementia): total (n) | Mean age of person with dementia | Meta-analysis | No. of included studies | No. of studies using tangible product of Life Story Work/Life Story Book | AMS-TAR overall score/ ENTREQ rating |
|---|---|---|---|---|---|---|---|---|---|---|---|
| 1 | Moos et al., 2006 | To describe the benefits of life story for nursing home residents with dementia. | Institutional care home | 1990–2003 | No restriction | Not reported | Not reported | N | 28 | 10 | Poor* |
| 2 | Subramaniam & Woods, 2012 | Focused on the potential psychosocial benefits of individualised reminiscence therapy for persons with dementia. | Nursing home | 2004–2010 | RCT | 90 | Mean age varied: 60–99 | N | 5 | 3 | 5 |
| 3 | Kindell et al., 2014 | Focused on review documents available for life story resources in dementia care. | Not reported | Not reported | Practical written resource of life story | Not applicable | Not applicable | N | 11 | 4 | Good* |
| 4 | Grøndahl et al., 2017 | Explored and described the use of life stories and its influence on persons with dementia in nursing homes, their relatives, and staff. | Nursing home | 2006–2015 | Pilot studies, multiple case studies, RCT | Persons with dementia, relatives and members or staff. | Not reported | N | 6 | 6 | 8 |
| 5 | Elfrink et al., 2018 | Provided an overview of how Life Story Books are used in dementia care. | Care home, home, both | 2003–2017 | Multiple baseline design, multiple qualitative case studies, qualitative studies, RCT | 243 | Mean age 58.6–92.8 years | N | 14 | 14 | 7 |
| 6 | Parker et al., 2020 | To critically appraise and uncover theories of change and outcomes for Life Story Work in dementia care to inform a feasibility study in a systematic review. | Nursing / care home, residential care, memory clinic, own home. | 1984–2014 | Not reported | 184 | Not reported | N | 16 | Not reported | 5 |

*(Continued)*

**Table 2.** (Continued)

| No | Authors / Year | Aim | Setting | Year of primary studies included | Study design of included studies | Population (person with dementia): total (n) | Mean age of person with dementia | Meta-analysis | No. of included studies | No. of studies using tangible product of Life Story Work/Life Story Book | AMS-TAR overall score/ ENTREQ rating |
|----|---------------|-----|---------|----------------------------------|----------------------------------|----------------------------------------------|----------------------------------|---------------|-------------------------|--------------------------------------------------------------------------|-------------------------------------|
| 7 | Doran et al., 2019 | To systematically review the literature regarding the experience of older people, families and staff using life-story work | Residential care facilities for older people | 2008–2016 | RCT, quasi-experimental, qualitative narrative, exploratory, multiple case studies | 150 | Not reported | N | 13 | 5 | 9 |

*Qualitative Review articles, rated based on ENTREQ criteria

dementia, except for studies by Doran et al. [32] where two of the primary studies included did not specify the diagnosis of the participants involved. After verifying the total number of persons with dementia in each study, at least 737 participants were included as two of the studies did not report the population involved. In these reviews, literature searches were performed from 1984 to 2017. There was a range of different settings involved: 30 studies were conducted at nursing homes, 43 studies at residential cares, 11 studies in participant's homes, one study at a memory clinic and 14 studies that did not specify the setting.

The AMSTAR evaluation is summarised in S1 Table. Three studies were rated moderate quality (score 4–7) and two studies were rated high quality (score 8–11) (S1 Table). ENTREQ summarised statement was used as a guideline in assessing the methodological quality of a qualitative review. When judged against the ENTREQ statement for studies by Moos et al. [33], the studies were methodologically poor due to the inadequate information for research questions, inclusion criteria, study screening, appraisal and data not being synthesised appropriately. Kindell et al. [14], however, showed good methodological quality with adequate and appropriate information provided. The appraisal and synthesis of the findings, however, were unclear. The summarised ENTREQ statement is reflected in the S2 Table.

## Characteristics and standard protocol in using the Life Story Book

Out of 93 primary studies included in this systematic review of reviews, 42 studies reported having a tangible product of a Life Story Book, with one study that did not specify the type of Life Story Book used. Table 3 summarises the characteristics and protocols for implementation of the Life Story Book in each of the included studies. Use of the Life Story Book varied and ranged from the traditional Life Story Book [10, 13, 14, 30, 32–34], memory book [33, 35], life history collage [13, 32], biographical history [13, 33], pen picture [13, 30], digital application of life story [30] and a rummage box [30, 36]. A wide variety of materials were used in creating the Life Story Book such as photographs, memorabilia, stories, narration, own words/quotations, favourite songs, and music [37]. The Life Story Books by Elfrink et al. [30] depicted the life stories of the individuals that were arranged in chronological order and illustrated with photographs and captions from childhood until the present. However, other studies reported the Life Story Book arranged according to topics of interest [14] with the person with dementia compiling it in the way they wish to narrate the story, with assistance from family members

**Table 3. Summary of characteristics and implementation of Life Story Book.**

| Author (year) | Characteristics of Life Story Book | | | | Who delivered Life Story Book? | Process of creating Life Story Book | Session time/ frequency | Length of Intervention |
|---|---|---|---|---|---|---|---|---|
| | Type | Materials used | Order of memories | Pages/no of memories | | | | |
| Moos et al., 2006 | Personal photographs, memory books, memory aids, biographical history, nostalgic videos and songs, paintings, and multi-sensory materials. | Personal photos, family biography, memorabilia, songs, and audiotapes of memories | Not reported | Not reported | Researchers, senior care assistants, dyads of persons with dementia and therapists. | Not reported | Varied: daily -weekly | Varied: 5 minutes—1 hour |
| Subramaniam & Woods, 2012 | Life story product: tangible Life Story Book, geriatric network kit, and general reminiscence conversations | Personal choice: picture, props, and words. | Personal preference | Not reported | Trained care-assistant / research team / social worker, and trainee clinical psychologist | Life review process, general reminiscence approach, and life phase discussions | Varied: 30 minutes—1 hour per week | Varied: 6–12 meetings in a 4–12-week period. |
| Kindell et al., 2014 | Life story product: Life Story Book | Information and photos | Chronological or topic of interest | Not reported | Care staff, carers, and volunteers | Not reported | Not reported | Not reported |
| Grøndahl et al., 2017 | Life story product: life history collage, Life Story Biography, tangible Life Story Book/pen, and picture | Pictures, photos, information about family/ careers, likes/ dislikes, and captions in own words | Not reported | Childhood, adulthood, family, and home. | Not reported | The Life Review and Experiencing Form (LREF), Family Biography Workshop, 6 key themes of Life Story Work | Not reported | Varied: 5–12 sessions in 4–12 weeks |
| Elfrink et al., 2018 | Life story product: tangible Life Story Book, digital book, rummage box, pen, pictures, and digital application. | Photographs, music, narration, stories, blank pages, quotations, and news | Chronological order | 2 pages (pen, picture)—70 pages; average length of movies: 18 minutes. | Person with dementia, partner/relative | The Life Review and Experiencing Form (LREF), Couples of life story approach. | Varied: 15–120 minutes | Varied: 3–16 sessions across 9–84 days |
| Parker et al., 2020 | Not reported | Not reported | Not reported | Not reported | Healthcare professionals, nursing care staff, SLPs, family members, researchers, and assistants | Not reported | Not reported | Not reported |
| Doran et al., 2019 | Life story product: life history narrative, life storytelling/ recording, life story collage, family bibliography, objects from life, life review sessions and Life Story Book | Not reported | Not reported | Not reported | Trained care/ nursing staff | Not reported | Not reported | Not reported |

and carers [10]. The number of pages in the Life Story Book varied from between two pages (pen picture) to 70 pages and the duration of the digital life story (which then was converted to a life story movie) ranged from 12 minutes to 27 minutes with an average of 18 minutes [30]. The Life Story Work reported by Kindell et al. [14] was an informal activity that the person with dementia could use to engage with relatives or carers while also having the potential to be a formal intervention. The Life Story Book session was reported as being delivered by the

researchers, caregivers, nurses, family members, relatives, and dyads of volunteers and thera-pists. Only some studies reported on the provision of training and supervision. Three studies used the structured Life Review and Experiencing Form (LREF) by Haight et al. [38] while oth-ers reported using the Couples of Life Story approach [30], Family Biography Workshop [39] and General Reminiscence approach [34, 40]. The actual use of the Life Story Book in terms of the number of sessions, frequency and duration of the session varied across the studies included in this review and is reflected in Table 3.

## Benefits of the Life Story Book for persons with dementia

The outcomes and benefits of using Life Story Books for persons with dementia are presented in Table 4. The positive outcomes reported include improvement in autobiographical mem-ory, mood, cognition, communication [10, 13, 30–32], quality of life [13, 30, 33], increased sense of self-integrity, identity, well-being, pride. and self-esteem that will eventually lead to changes to better behaviour of the person with dementia [31–33].

## Effects of using the Life Story Book on relatives

The effects of using Life Story Books on the relatives, caregivers, and care staff of persons with dementia are presented in Table 4. Most of the reviews reported that the Life Story Book plays an important role in aiding communication. The improved quantity and quality of verbal interaction between a person with dementia and the carer reported by Moos et al. [33] are in

**Table 4. Summary of the benefits and effects of Life Story Book in dementia care.**

| Author, year | Outcome for person with dementia | Effect of Life Story Book on the relatives/ caregiver/ care staff | Communication |
|---|---|---|---|
| Moos et al., 2006 | Enhanced quantity and quality of verbal interaction, self-integration, sociability, quality of life and sense of well-being and self-esteem. | Staff-reported increased motivation in care and interactive communication. | Improved quantity and quality, increased social engagement and interaction. |
| Subramaniam & Woods, 2012 | Positive well-being, autobiographical memory, mood, communication, and cognition. Supported and promoted personhood, reduction in depression, disorientation, fear, and anxiety. | Caregiver-rated improvement in mood, decreased burden and behaviour problems. | Reported enhanced communication without details of the type of interaction and conversation. |
| Kindell et al., 2014 | NI | NI | NI |
| Grøndahl et al., 2017 | Cognition, autobiographical memory, depression, positive mood, communication, and quality of life. | Relative-reported stimulation of memories, positive behaviour points-of-reference for communication for staff. | Facilitated communication. |
| Elfrink et al., 2018 | Improved autobiographical memory, cognitive functioning, mood, depression, quality of life, quality of relationships, and communication with caregiver. No significant improvement in independence, memory, or behaviour problems. | Care-staff and relative-reported improvements in relationships, partner affirmation, engagement, fullness of life as a couple, enhanced communication, and social interactions. | Reported improvements in communication and social interactions. |
| Parker et al., 2020 | Increased self-worth, reduced anxiety, depression, agitation, mood, negative behaviour, enhanced interactions, and communication. | Carers and staff-reported improvements in relationships and involvement in care-planning and delivery, allowed effective engagement of family members/carers. | Increased communication skills |
| Doran et al., 2019 | Maintained self-identity, built, and maintained relationships, aided quality interactions, and prompted communication in severe cognitive impairment. | Staff reported better provision of care by staff, enabled the organisation of suitable activities, built, and maintained relationships through better communication helped the family with care process, and improved strategies to overcome aggressive behaviour. | Facilitated communication and stimulated interactions between persons with dementia, family members, residents, and staff. |

NI = No information

line with the other reviews that revealed that the Life Story Book was proven to enhance interaction and promote points of reference to communicate to the staff and relatives [10, 13, 30–32]. The effect of enhanced communication meaningfully helps in maintaining and building the relationship between persons with dementia and their family, relatives, and caregivers. The relationship creates an effective engagement between persons with dementia and the carer that eventually promotes better care of person-centred care towards persons with dementia. In addition, staff and carers reported that Life Story Book helps them to understand and have more knowledge about the person with dementia [13, 30] and enable them to alter their strategies to overcome aggressive behaviour more effectively [31, 32].

## Discussion

The Life Story Book is one of the products of reminiscence therapy that can be used to enhance person-centred care among persons with dementia [10]. However, across the studies reviewed, there is no single approach or standard definition of 'life story work' [41]. The definition appears to be varied between the approaches of life story work depending on the aim and objective of the life story work itself. This is consistent with the reviews by Kindell et al. [14] and Doran et al. [32] as to whether Life Story Book should be used as an informal activity for persons with dementia to be engaged in or as a formal intervention run by trained staff or therapists.

There is no consensus found in terms of the characteristics or standard procedure of creating a Life Story Book such as topics to be included, the order of memories, the number of pages or how it should be used in a session. Thus, this study suggests that creating an individualised Life Story Book for persons with dementia that caters for their aims and goals would maximise the benefits of using the book. It also allows healthcare professionals in a dementia care setting to work around goals according to each person with dementia's cognitive and communication demands. With regards to the duration of each session, most studies reported that sessions lasted between 30 minutes to an hour, based on the condition of the persons with dementia. Ingersoll-Dayton et al. [42] proposed that sessions were most productive when conducted weekly. This could provide SLPs and other dementia care professionals with insights into the length and frequency of sessions that are adequate for effective intervention.

Haight et al. [38] suggested that the tangible Life Story Book should be based on the outcome of the life review process. Kindell et al. [14] suggest that this intervention is reliant on the person who completes the Life Story Book. Similar recommendations were stated by Subramaniam and Woods [10] in which the Life Story Book should be created by the person with dementia using their own choices of pictures, props, and words, with the order depending on how they would personally like to narrate their life stories. This may be an issue, however, for the person living with dementia who has impaired memory and is unable to recall recent events or pleasurable times. These persons may not have the capacity to maintain a sense of continuity between events and memories [12]. More flexibility and naturalistic processes are required according to the person's cognitive and emotional needs, as well as preferences and progress of dementia [10, 41]. In that case, the involvement of a family member or caregiver is required to aid the process of creating a Life Story Book. Findings from Morgan and Woods [43] suggest that collaborating with family and/or relatives in developing the Life Story Book for their loved ones is indeed effective. Family and/or relatives of persons with dementia are encouraged to actively participate in the process of selecting materials and designing the Life Story Book alongside persons with dementia as sometimes they have more knowledge and information about the person with dementia's habits and preferences.

Despite that, promoting self-identity, self-value, improvement in social communication, and creating and building relationships between family, relatives, and care staff, have been highlighted across all the reviews studied. The Life Story Book focuses on enabling the person with dementia to share and talk about their life stories and experiences [18, 28] and maintain their identity. This is similar to the findings of McKeown et al. [12] that a Life Story Book helps to reinforce the sense of identity, self-esteem and pride of the person with dementia. As a consequence, the family, relatives and staff develop insight to see the person with dementia as a whole person and recognise their unique identity, history, interest and their whole life [12, 28]. This could further help family, relatives, carers, and health practitioners to organise activities or exercises that are beneficial for the person living with dementia.

Another notable benefit of the Life Story Book is enhanced communication among persons with dementia. The study found that the Life Story Book acts as a valuable tool in aiding communication between persons with dementia and their family, relatives, and care staff. One of the theories of the Life Story Book outlined by Parker et al. [31] is that a Life Story Book helps others to get to know the person with dementia better, which eventually improves the interaction and communication between the person with dementia and the staff. This is consistent with the literature findings where the staff perceived the Life Story Book as a point of common reference to communicate with persons with dementia and becomes a 'kit' as a prompt to aid the communication [13, 44]. As a result, fewer incidences of negative behaviour and mood changes can be observed among persons with dementia as they can express themselves better. However, quantitative findings on how the Life Story Book was used to enhance both verbal and non-verbal interactions and communication are scant and requires further research.

Utilising a Life Story Book in dementia care supports the person with dementia to socialise and feel accepted as a part of the social network in their current surroundings and facilities [45]. Subramaniam et al. [15] reported that improvement in the relationship between the person with dementia and caregivers was observed right after the Life Story Book was introduced to them. The improvement in communication and enhanced interaction between them could be the reason behind it [46]. This creates a better care practice, a better quality of life and a warm relationship due to the enhanced relationship between the family, care staff, and the person with dementia itself [15]. However, the studies in this review mostly reported the engagement and improvement of the relationship between persons with dementia and their caregivers and staff in nursing or residential and care home settings [10, 13, 30, 33]. Similar findings were stated by Gridley et al. [47], where Life Story Book was widely used in hospitals and care home settings and less in a home setting. Greater insights and benefits may have come from a focus on the engagement and bonding of the close family members and relatives in their home setting which they are more familiar with.

Moreover, the use of a Life Story Book is beneficial for SLPs when providing effective intervention for persons with dementia. From the present study, it is evident that the Life Story Book promotes self-empowerment and improves communication skills and memory among persons with dementia. The service delivery of SLPs and other healthcare professionals who work closely with persons with dementia could focus on a wide range of communication and cognitive goals with the aid of the Life Story Book. For instance, healthcare professionals could work on goals of promoting social participation, reducing frequencies of responsive behaviours, and facilitating daily living activities [48] with the individuals and their families or caregivers. Thus, SLPs could use a Life Story Book as an effective intervention during their clinical sessions with persons with dementia. This is further supported by Bourgeois [24], where the researcher encouraged more professionals to use Life Story Book as a tangible product to provide better care for them.

Apart from the benefits of Life Story Book, McKeown et al. [12] identified a few challenges in using Life Story Book. First, in terms of the content of the Life Story Book, it is challenging for healthcare professionals when sensitive information and personal disclosures emerge and the issues of upsetting memories are brought up in the life review process [14]. Having a person with dementia tell their own stories could be a challenge with the nature of their inability to recall memories [12]. The Life Story Book might be overused or underused in dementia care due to lack of time and is identified to be one of the challenges among caregivers [49]. Thus, finding the right balance is important. Other barriers to using the Life Story Book as an intervention tool are the lack of a standard format for collecting information and the lack of available suitable resources to develop a personalised Life Story Book [12]. It should be highlighted that most of the staff and carers have limited knowledge and skills towards dementia care, especially in using the Life Story Book. According to previous research, the staff of nursing homes faced major difficulties in delivering high-quality care due to a lack of professional health education [13]. Therefore, more guidance and training should be provided on how to use the Life Story Book to the SLPs, families, relatives, care staff or individuals involved in delivering services for persons with dementia [47]. This is crucial to support the implementation of this practice in dementia care and to ensure the best outcomes of the intervention [21, 47].

## Limitations

The search was limited to English language sources and relevant sources in other languages may contain useful information that may have been excluded in this systematic review. This systematic review incorporated review studies that aimed to explore different objectives in each study. The available review studies included in this study did not portray a consensus on the findings, especially in terms of the characteristics and implementation of the Life Story Book. It was therefore a challenge to compare the studies and arrive at a consensus. Despite these limitations, this review provides a comprehensive overview of the usage of life stories in dementia care.

## Implications

The Life Story Book assists in supporting dementia care and is a beneficial activity to the person with dementia, staff, nurses, families, relatives, and caregivers. Life Story Book has become a form of evidence-based practice and should be recognised by healthcare service providers including SLPs, nurses, care staff and managers when planning services for dementia. It is recommended that SLPs use the Life Story Book in planning the treatment as it is evident that the Life Story Book enhances communication and increases interactions when supplemented with communication strategies to elicit more meaningful conversations. Challenges that may arise highlight the importance and need for careful planning, education, and support in the implementation of the Life Story Book in practice. Future research to explore the use of the Life Story Book is recommended for enhancing communication, the value of verbal and non-verbal interactions and the social interaction itself in direct communication intervention and in the long term.

## Conclusion

The Life Story Book is the product of a personalised collection of life events and memories that promotes person-centred care in dementia management. This systematic review collected and synthesised findings from a review of the usage of life stories among the dementia population. The challenges faced have been outlined to ensure better implementation of strategies while

delivering the service. Some positive outcomes for the person living with dementia have been reported, although more robust research is needed to determine the effects of the Life Story Book for this population. There is, undoubtedly, a need to measure the impact and influences of the Life Story Book towards the caregiver, family, and relatives.

## Supporting information

**S1 Table. Quality assessment ratings of review studies included (AMSTAR).**
(DOCX)

**S2 Table. ENTREQ methodology quality assessment for qualitative review studies.**
(DOCX)

**S1 Checklist. PRISMA 2020 for abstracts checklist.**
(DOCX)

**S2 Checklist. PRISMA 2020 checklist.**
(DOCX)

## Author Contributions

**Conceptualization:** Ponnusamy Subramaniam, Preyaangka Thillainathan, Nur Amirah Mat Ghani, Shobha Sharma.

**Data curation:** Preyaangka Thillainathan, Nur Amirah Mat Ghani.

**Supervision:** Ponnusamy Subramaniam, Shobha Sharma.

**Writing – original draft:** Preyaangka Thillainathan, Nur Amirah Mat Ghani.

**Writing – review & editing:** Ponnusamy Subramaniam, Shobha Sharma.

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
