## [Decision Letter · Decision Letter 0]

21 Jul 2023

PONE-D-22-30927Life Story Book to enhance communication in persons with dementia: A systematic review of reviewsPLOS ONE

Dear Dr. Sharma,

Thank you for submitting your manuscript to PLOS ONE. After careful consideration, we feel that it has merit but does not fully meet PLOS ONE’s publication criteria as it currently stands. Therefore, we invite you to submit a revised version of the manuscript that addresses the points raised during the review process.

We look forward to receiving your revised manuscript.

Kind regards,

Michael Thomas Lawless, Ph.D.

Academic Editor

PLOS ONE

Journal Requirements:

Additional Editor Comments (if provided):

We thank the authors for the opportunity to consider this manuscript for publication. The reviewers had positive comments about the approach to the literature review. Reviewer 2 praises the discussion as excellent. However, the reviewers have also identified areas for development. Please see the reviewers' comments and consider revising the manuscript by addressing them. If you choose to do so, please pay attention to grammar, terminology, and punctuation, impacting the clarity and precision of the content. Tables need improvement for readability, and consistent terminology usage is required.

Reviewers' comments:

Reviewer's Responses to Questions

**Comments to the Author**

1. Is the manuscript technically sound, and do the data support the conclusions?

Reviewer #1: Partly

Reviewer #2: Partly

2. Has the statistical analysis been performed appropriately and rigorously? 

Reviewer #1: N/A

Reviewer #2: N/A

3. Have the authors made all data underlying the findings in their manuscript fully available?

Reviewer #1: Yes

Reviewer #2: Yes

4. Is the manuscript presented in an intelligible fashion and written in standard English?

Reviewer #1: Yes

Reviewer #2: Yes

5. Review Comments to the Author

Reviewer #1: The approach to the literature review is sound, as it includes relevant approaches for the different study types selected.

While the conclusions are supported by the evidence presented, there are some issues that need addressing, as follows:

1. Evidence - unless there is definitive evidence that the life story book actually improves quality of care, interactions, communication, memory, quality of life etc., then these cannot be claimed. It appears that most of the evidence relied on is anecdotal, i.e. what the staff or family report from their own observation, rather than from clinical assessment or direct observation by a researcher. On Page 3, lines 118-120, i suggest the statement be changed to '...quality of life and other outcomes for the person living with dementia. Benefits may include improved cognition and mood, reduced depression...'. It is important not to overstate the benefits of the life story book.

Wherever terms such as 'significant' (Conclusion) are used, replace these with more suitable terms.

2. Grammar - there are many areas where incorrect grammar is used and this makes it hard to understand what message is conveyed, or it invalidates the statement. I have listed recommendations on use of correct grammar, as follows:

Abstract:

2nd sentence - replace 2nd sentence with 'This involves collecting the life stories and memories of the person living with dementia and compiling them into a book or folder, which is used by staff or family to assist the person recall these memories'.

3rd sentence - replace the 3rd sentence with 'Evidence on the use, benefits and influences of the life story book in dementia carte is limited.' Commence another sentence as 'This systematic literature review aimed to ...'.

Last sentence - This needs to be broken into two sentences, finishing at 'communication'. Commence a new sentence as 'Guidelines and training are also required to make the best use of the life story book'.

Page 2, line 92. Replace 'nurses that' with 'nurses who'.

Page 3, lines 98-99. Replace 'better delivery of care for the person with dementia that it becomes more person centred care as a result' with 'more person-centred care for persons living with dementia'.

Page 3, lines 118-120. Replace 'quality of care' with 'quality of life', as care quality does not support the following statement. i suggest the statement be changed to '...quality of life and other outcomes for the person living with dementia. Benefits may include improved cognition and mood, reduced depression...'. It is important not to overstate the benefits of the life story book.

Page 9, line 238. There are missing words at the beginning of the sentence, commencing '[14]'.

Page 17, line 246. Replace 'The life story book product used varied..' with 'Use of the life story book varied for ..'

Page 18, line 266. Replace '..three of the reviews reported studies used the...' with 'Three studies used the ...'.

3. Terminology - many terms are used inappropriately, which alters the true meaning of what is being conveyed. I have listed recommended words to replace ones used incorrectly, as follows:

Page 2, line 91. Replace 'vastly implemented' with 'widely implemented'.

Page 2, line 95. Replace 'that states' with 'that indicates'.

Page 7, line 179. Replace 'reviews' with 'review'.

Page 8, line 199. Use a more suitable term to replace 'database hits'.

4. Punctuation - there are places where commas need to be included, as follows:

Page 17, line 247. Insert comma between reference [33] and 'memory book'.

Page 17, line 248. Insert comma between reference [30] and 'digital'.

5. Findings.

Page 9. regarding reference to results of authors Moos et al [33] and some other author [14], it would be preferable to report these results with all other results in the most relevant place. Include the Author of ref [14] in this sentence.

Table 3. Remove all superfluous words to ensure the content is precise and consistently stated, e.g. for ref Moos et al., 2006. Colum titled 'Type' remove words 'this study' and any other descriptive statements.

Further review comments - see attached file.

Reviewer #2: I've gone through and suggested some changes closer to academic prose. Your discussion is excellent; your spacing makes the tables VERY hard to read, which is why I chose 'partly' above. Please make the tables more standard in font size and spacing.

6. PLOS authors have the option to publish the peer review history of their article (what does this mean?). If published, this will include your full peer review and any attached files.

Reviewer #1: No

Reviewer #2: No

---

## [Author Response · Author response to Decision Letter 0]

30 Jul 2023

Dear respected Dr Lawless,

We are immensely grateful to be given the opportunity to submit our revised manuscript, Life Story Book to enhance communication in persons with dementia: A systematic review of reviews, to PLOS ONE. We would also like to thank the Reviewers for guiding us almost every step of the way to make our manuscript worthy of publication.

The revised version addresses each of the the points raised by the Reviewers during the review process, and we hope that the corrections we have made comply with the requirements for publication. Detailed responses and corrections made to the revised manuscript submission have been uploaded as a separate file "Response to Reviewers LSB 30072023".

We confirm that this work is original and has not been published elsewhere, nor is it currently under consideration for publication elsewhere.

Thank you once again for your kind consideration of this revised manuscript.

Sincerely,

Shobha Sharma

(Corresponding Author)

---

## [Decision Letter · Decision Letter 1]

4 Sep 2023

Life Story Book to enhance communication in persons with dementia: A systematic review of reviews

PONE-D-22-30927R1

Dear Dr. Sharma,

We’re pleased to inform you that your manuscript has been judged scientifically suitable for publication and will be formally accepted for publication once it meets all outstanding technical requirements.

Kind regards,

Michael Thomas Lawless, Ph.D.

Academic Editor

PLOS ONE

Additional Editor Comments (optional):

Reviewers' comments:

Reviewer's Responses to Questions

**Comments to the Author**

1. If the authors have adequately addressed your comments raised in a previous round of review and you feel that this manuscript is now acceptable for publication, you may indicate that here to bypass the “Comments to the Author” section, enter your conflict of interest statement in the “Confidential to Editor” section, and submit your "Accept" recommendation.

Reviewer #2: All comments have been addressed

Reviewer #3: All comments have been addressed

2. Is the manuscript technically sound, and do the data support the conclusions?

Reviewer #2: Yes

Reviewer #3: Yes

3. Has the statistical analysis been performed appropriately and rigorously? 

Reviewer #2: N/A

Reviewer #3: N/A

4. Have the authors made all data underlying the findings in their manuscript fully available?

Reviewer #2: Yes

Reviewer #3: Yes

5. Is the manuscript presented in an intelligible fashion and written in standard English?

Reviewer #2: Yes

Reviewer #3: Yes

6. Review Comments to the Author

Reviewer #2: Greatly improved. The two appendices are highly useful. The English is stronger and better organized as well.

Reviewer #3: (No Response)

7. PLOS authors have the option to publish the peer review history of their article (what does this mean?). If published, this will include your full peer review and any attached files.

Reviewer #2: No

Reviewer #3: No

---

## [Editor Report · Acceptance letter]

25 Sep 2023

PONE-D-22-30927R1 

Life Story Book to enhance communication in persons with dementia: A systematic review of reviews 

Dear Dr. Sharma:

I'm pleased to inform you that your manuscript has been deemed suitable for publication in PLOS ONE. Congratulations! Your manuscript is now with our production department. 

Kind regards, 

on behalf of

Dr. Michael Thomas Lawless 

Academic Editor

PLOS ONE